# Evaluation of the Polysaccharide “Immeran” Activity in *Syrian hamsters’* Model of SARS-CoV-2

**DOI:** 10.3390/v16030423

**Published:** 2024-03-09

**Authors:** Liubov Viktorovna Generalova, Denis Pavlovich Laryushkin, Irina Anatolievna Leneva, Anna Valerievna Ivanina, Galina Vladimirovna Trunova, Sergei Vladimirovich Dolinnyi, Evgenii Aleksandrovich Generalov

**Affiliations:** 1Faculty of Medicine, Peoples’ Friendship University of Russia (RUDN University), 117198 Moscow, Russia; generals1100@mail.ru (L.V.G.); sdolinny.ru@yandex.ru (S.V.D.); 2Federal Research Center “Pushchino Scientific Center for Biological Research of the Russian Academy of Sciences”, Institute of Cell Biophysics of the Russian Academy of Sciences, 142290 Pushchino, Russia; mr.ldp@yandex.ru; 3Mechnikov Research Institute of Vaccines and Sera, Department of Virology, 105064 Moscow, Russia; wnyfd385@yandex.ru (I.A.L.); ivanina.anna97@mail.ru (A.V.I.); 4Federal State Budgetary Institution National Medical Research Radiological Center (FSBI NMRRC) of the Ministry of Health of the Russian Federation, P.A. Hertsen Moscow, Oncology Research Institute, 125284 Moscow, Russia; gtrunovamnioi@mail.ru; 5Faculty of Physics, Lomonosov Moscow State University, 119991 Moscow, Russia

**Keywords:** Immeran, polysaccharide, *Solanum tuberosum* L., SARS-CoV-2, COVID-19, *Syrian hamsters*, respiratory disease

## Abstract

COVID-19 is a highly contagious respiratory disease with a high number of lethal cases in humans, which causes the need to search for new therapeutic agents. Polysaccharides could be one of the prospective types of molecules with a large variety of biological activities, especially antiviral. The aim of this work was to study the specific antiviral activity of the drug “Immeran” on a model of a new coronavirus infection SARS-CoV-2 in hamsters. Based on the second experiment, intraperitoneal treatment with the drug according to a treatment regimen in doses of 500 and 1000 μg/kg (administration after an hour, then once a day every other day, a total of 3 administrations) was effective, reliably suppressing the replication of the virus in the lungs and, at a dose of 1000 μg/kg, prevented weight loss in animals. In all cases, the treatment stimulated the formation of virus-neutralizing antibodies to the SARS-CoV-2 virus, which suggests that the drug possesses adjuvant properties.

## 1. Introduction

In December of 2019, an outbreak of SARS was reported in the most populous city in central China (Wuhan). The cause of the disease was a previously unknown type of coronavirus. The isolated strain was named 2019-nCoV. The Center for Disease Control and Prevention of the People’s Republic of China (CDC), in January, deciphered and published the complete genome of the isolated virus on the GenBank portal. Phylogenetic analysis has shown that this is a new coronavirus, genetically closest to the coronaviruses associated with severe acute respiratory syndrome (SARS), and closely related to the genome of the SARS-like coronavirus of bats, with a genomic organization typical of the betacoronavirus genus. The International Committee on Taxonomy of Viruses assigned the official taxonomic name SARS-CoV-2 to the new coronavirus and COVID-19 to the disease [1,2,3]. Within a short time, the virus from China spread across countries and continents. At the beginning of March 2020, WHO declared a COVID-19 pandemic. Many new strains of this virus appeared and who knows how many more of them are coming [4].

The World Health Organization recommends vaccination as the main means of combating the new coronavirus infection [5]; however, its effectiveness is limited due to the rapid variability of circulating viruses [6]. In addition, any vaccine is directed against a specific target and, therefore, in addition to vaccination, WHO recommends the use of antiviral drugs, which have their own limitations of applicability [7,8,9].

There are several antiviral drugs [10], including favipiravir, which are approved for the treatment of viral diseases [11] and have been studied in clinical trials in COVID-19 patients [12,13,14,15]. However, there are some works questioning the efficacy of standard approved drugs in COVID-19 patients—favipiravir [14,16], remdesivir [17], interferon therapy [18], hydroxychloroquine [19] and lopinavir [20]; at the same time, adverse side effects of these drugs in COVID-19 patients were exhibited to a full extent.

One of the promising directions in the search for new therapeutic agents for viral diseases are polysaccharides, which have a whole range of useful properties from the point of view of the biomedical industry. The antiviral properties of natural polysaccharides have been known for a long time and there are even drugs based on them, such as glycosaminoglycan and chitosan [21]. For example, polysaccharide from Chaga effectively inhibits RNA viruses in an in vitro model (*Caliciviridae*, *Coronaviridae* и *Orthomyxoviridae*) and DNA-viruses (*Alphaherpesvirinae* and *Parvovirus*) [22]. Polysaccharide from *Helianthus tuberosus* L. demonstrates antiviral activity in vitro and in vivo models, including murine herpetic meningoencephalitis, due to its interferon-inducing and immunomodulatory properties [23,24]. It is well known that most polysaccharides have a high affinity for the receptors of immunocompetent cells (TLR, CLEC, NOD, SR and others), which are the basis of their biological activity [25,26]. Correct immunoregulation during viral SARS-CoV-2 attack could be a possible mechanism for the effective treatment, prevention and aggravation of the COVID-19 disease [27,28,29].

In order to delimit the spread of the virus and create a reserve of necessary drugs, the entire medical community is focusing on the use of already existing registered drugs, which include the polysaccharide drug “Immeran”, extracted from the shoots of *Solanum tuberosum* L. [30].

## 2. Results

### 2.1. Effect of “Immeran” on SARS-CoV-2 Replication in the Hamster Model (First Series of Experiments)

We first examined whether “Immeran”, which is currently approved for ulcer disease, has inhibitory effects on SARS-CoV-2 replication in *Syrian hamsters*. In the first series of experiments, we studied the effectiveness of various schemes and doses of drug administration. The effectiveness was assessed by determining the virus isolated from the lungs of experimental hamsters, analyzing changes in the body weight of the animals, as well as studying pathological changes in the lungs through histological examination.

In the first series of experiments, the virus titer in the lungs of hamsters in the control group on day 5 after infection was high, up to 6.50 ± 0.84 lgTCID_50_/mL. A single prophylactic use of the drug in animals, started 24 h before infection, at both doses of 250 and 500 μg/kg, did not reduce the virus titer in the lungs; it was practically no different from that in the viral control group. Therapeutic use of the drug, started 1 h after infection and continued for 4 days at a single dose of 250 μg/kg, reduced the titer of the virus, but this decrease was not statistically significant (5.80 ± 1.48 lgTCID_50_/mL). At the same time, the increased dose of the drug—500 μg/kg (Group 5)—led to almost complete suppression of virus replication in the lungs (Figure 1).

The use of favipiravir (Group 6) was not effective and did not affect virus titer in the lungs of animals (6.60 ± 1.09 lgTCID_50_/mL).

During the experiment, until the animals were euthanized, there was no weight loss observed in “Immeran” and viral control groups. At the same time, in the group of animals treated with favipiravir, weight loss was observed, reaching up to 5% on the second day after infection (Figure 2).

### 2.2. Lung Tissue Histology after Treatment

In both series of the experiments, in all animals of Group 1, the morphological picture of inflammatory changes in the lungs corresponded to lobar viral pneumonia (Figure 3).

The histological study of the golden hamsters’ lungs from the viral control group (group 1) showed that, on the 4th day after infection, pronounced alterative-inflammatory changes developed in the airways and respiratory tract of all animals, the morphological picture of which corresponded to the interstitial bronchopneumonia. In the lobes of the right lungs, extensive confluent airless foci of pneumonia were revealed (Figure 3a).

Compared to the viral (group 1) and Favipiravir control (group 6) groups, the various treatment regimens with the drug “Immeran” in groups 2, 3, 4 and 5 led to a progressive decrease in the area and severity of inflammatory changes in the airways and respiratory part of the lungs (Table 1, Figure 3b–d,f,g).

In Groups 2–4, the morphological picture corresponded to viral pneumonia. The nature of inflammatory changes in the lungs of the animals did not differ from that in animals from viral control. In Group 3, in 4 animals out of 5, small scattered airless foci and adjacent areas with reduced airiness were noted, the area of which, in total, ranged from 1–5% to 20% (up to 10 times less than in the control group); in the lungs of hamster No 2, there were no inflammatory changes.

In Group 4, the area of the lung parenchyma with inflammatory changes in these animals was small, no more than ¼ of the total section area (<1% of the lung). In animals № 1 and 4, the histostructure of the lungs was normal. In Group 5, only 2 out of 5 animals had signs of viral pneumonia. In No. 3, 4 and 5, no deviations from the norm were detected. In Group 6 with the favipiravir treatment, the morphological picture of which, in animals No 2 and 3, corresponded to focal viral pneumonia in combination with aspiration bronchopneumonia, the foci of which were multiple and often bordered on foci of viral pneumonia. In hamsters No 1, 4 and 5, foci of alveolitis (1 and 4), and small confluent foci of viral pneumonia located in all the lobes of the lung, were noted. It should be emphasized that in the histological specimen of hamster lung No 5, obstructive organizing thrombi were identified in the lumen of the branches of the pulmonary artery accompanying the lobar bronchus and bronchus of the 2–3rd generation.

In a morphological study of the lungs of *Syrian hamsters* euthanized on the 5th day after infection, alterative inflammatory changes were determined that corresponded to viral pneumonia, the severity of which progressively decreased in Groups 3, 4 and 5, compared with the viral control group (Group 1) and Favipiravir control (Group 6).

### 2.3. Effect of “Immeran” on SARS-CoV-2 Replication in the Hamster Model (Second Series of Experiments)

The virus titer in the lungs of hamsters in the second experiment was also determined on day 5 after viral infection. In the control group of untreated animals (Group 1), the virus titer was 7.70 ± 0.50 lgTCID_50_/mL. Compared to the first experiment, in the prophylactic regimen, the dose was increased to 1000 μg/kg body weight (Group 2); this resulted in a decrease in the virus titer in the lungs of the animals, lower than in the viral control group (5.83 ± 0.88 lgTCID_50_/mL), unlike in the first experiment; however, this decrease was not statistically significant and needs to be evaluated using more animals. Groups 4 and 5 showed the reduction of the virus titer by more than 2 lgTCID_50_/mL (5.50 ± 1.24 lgTCID_50_/mL—Group 4, and 5.00 ± 0.69 lgTCID_50_/mL—Group 5).

In the second series of the experiments, weight loss in the viral control group (Group 1) was greatest on day 7 after infection, reaching up to 5%. From day 5, the animals began to steadily gain weight. Weight loss in the first days and weight gain in subsequent days in hamsters treated with the drug, according to the prophylactic regimen (Group 2) and the treatment regimen when administered every other day at a dose of 500 μg/mL (Group 4), was similar to that in the viral control group. At the same time, in Group 3, in which the suppression of the virus in the lungs was most significant, weight loss in animals was undetected. The data were obtained for Groups 4 and 5 (Figure 4).

Treatment with 500 μg/kg (Group 3) significantly suppressed the multiplication of the virus in the lungs of the animals (3.20 lgTCID_50_/mL) and protected them from weight loss (Figure 5).

### 2.4. Neutralization Reaction

Our studies revealed the presence of antibodies in a significant titer (above 1:10) to both strains of the virus in all infected animals. There was no evidence that the antibody titer in treated animals was lower than the antibody titer in untreated animals, indicating that the drug does not inhibit antibody formation. At the same time, it is important to note that in all the treated groups, the titer of antibodies to the “Dubrovka“ Wuhan-like strain with which animals were infected was higher than in the viral control group. Results show that “Immeran” has immunoadjuvant properties. At the same time, the highest titers of virus-neutralizing antibodies were formed in animals in Group 3 (up to 5-fold compared to the control), treatment in which, according to the results of the two experiments, was the most effective, significantly suppressing replication of the virus in lungs and completely preventing weight loss. In Group 6 (uninfected animals), no antibodies were detected. Results are shown in Table 2.

According to the second experiment, treatment with the drug intraperitoneally in doses of 500 and 1000 μg/kg (administration after an hour, then once a day every other day, a total of 3 administrations) was effective, significantly suppressed replication of the virus in lungs and, at a dose of 1000 μg/kg, prevented animals from losing weight.

A single prophylactic use of the drug in doses of 500 and 1000 μg/kg in a model of *Syrian hamsters* infected with SARS-CoV-2 was ineffective; the titer of the virus in lungs of the animals was practically that in the viral control group.

The use of “Immeran” in all dosages and regimens, including prophylactic, led to a decrease in the area of damage to the lung tissue in comparison with the control, up to the normal state of the lung tissue. In all cases, use of the drug did not have a suppressive effect on the formation of virus-neutralizing antibodies to SARS-CoV-2 virus. On the contrary, in all cases, the treatment stimulated the formation of antibodies, which could possibly give an opportunity to suggest the presence of adjuvant properties of the drug.

In groups treated with “Immeran”, both histological and virus titer data confirm the inhibition of viral replication in the lung tissue. According to the results of both experiments (“Dubrovka“ Wuhan-like and Omicron strains were used), the most effective treatment was that with the drug given intraperitoneally at a dose of 500 μg/kg (administration one hour after infection, then once a day for 4 days), which significantly suppressed the replication of the virus in the lungs, prevented weight loss in the animals and, according to histological studies, significantly improved condition of the lungs.

During daily clinical examination of the animals, no allergic or pathological reactions to the administration of “Immeran” were detected. Hyperemia, swelling of the skin and subcutaneous tissue was not observed, and the fur remained smooth, shiny, and its integrity was not compromised. Increased pain during the administration of the drug was also not detected.

## 3. Discussion

Polysaccharides are a heterogeneous group of biologically active molecules, which has different mechanisms of action. Some carbohydrates interact with receptors, some with the cellular membrane, some with different signaling and biologically active molecules. Polysaccharides with high molecular weight possess immunomodulatory activity in different ways and one of them is interaction with the extracellular matrix (ECM). Pappas, A.G. and colleagues showed that veriscan (a large chondroitin sulfate proteoglycan) resides in the ECM influencing vascular permeation, tumor homing and malignant pleural effusion, and enhancing tumor cell proliferation. On the other hand, a veriscan-deficient tumor lacks tumor-associated macrophages and neutrophils, and T-regulatory cells [31].

Complex molecules, such as peptidoglycans (PGs), interact with other immune-active molecules, such as chemokines and interleukins, by regulating the ionic strength of a solution and direct interaction with water molecules. Thus, PGs can regulate ligands that control intracellular signaling events and influence the inflammatory response. Also, it has been found that they act as signaling molecules activating immune-corresponding intracellular molecular pathways [32,33].

At the same time, well-known hyaluronan takes part in both the inter- and extracellular matrix. Johnson, P. with co-authors showed that this type of polysaccharide plays an immune role in the lungs through binding with the macrophage CD44 receptor and different TLRs. Such interactions lead to the regulation of inflammatory infiltrate, tissue repair processes and type I inflammatory response [32].

The data obtained suggest that the drug “Immeran” potentially has antiviral properties, and its reparatory characteristics could be useful in the case of lung tissue damage. At the same time, we found this polysaccharide’s ability to increase the quantity of virus-neutralizing antibodies. The highest titers of virus-neutralizing antibodies were detected in animals in Group 3; treatment was the most effective, significantly suppressing the replication of the virus in the lungs and completely preventing weight loss. 

In previous preliminary experiments, we found that “Immeran” had no direct antiviral activity against SARS-CoV-2 (“Dubrovka” Wuhan-like strain) in Vero cell culture. Thus, in the case of the coronavirus infection, “Immeran” can—during several stages of the pathological process—stimulate the regeneration of lung parenchyma, activate antiviral immune response and reduce the level of pro-inflammatory cytokines, which are crucial in tissue damage and in immune response. Moreover, this polysaccharide significantly stimulates both the number of spleen cells and lysis zones in an agar plate, thus stimulating antibody formation and being an immunomodulator, according to the Jerne–Nordin model [30].

The data obtained suggest that the drug “Immeran” may have antiviral properties and its reparatory characteristics could be useful in the treatment of lung tissue damage. We also found that this polysaccharide can stimulate the expression of virus-neutralizing antibodies. The highest titers were detected in Group 3 animals, where treatment in two experiments significantly suppressed viral replication in the lungs and prevented weight loss. 

Antiviral activity of “Immeran” has been studied in the *Syrian hamster*’s model of SARS-CoV-2 infection. The most effective treatment was “Immeran” given intraperitoneally at a dose of 500 μg/kg one hour after infection, then once a day for 4 days. It was found that “Immeran” significantly suppressed replication of the virus in lungs, prevented weight loss in the animals and significantly improved the condition of the lungs.

We acknowledge the need for improvement in the presentation of the antiviral effects of “Immeran.” In previous preliminary experiments, we observed no direct antiviral activity of “Immeran” against SARS-CoV-2 in vitro. However, in the context of the coronavirus infection, “Immeran” exhibited significant antiviral actions, including the suppression of viral replication in the lungs, prevention of weight loss, and improvement of lung condition, as demonstrated in the Syrian hamster model.

Our study has some limitations. SARS-CoV-2 infection in *Syrian hamsters* is widely regarded as the most representative of the human disease, making this model pivotal for assessing drug antiviral efficacy. However, like any animal model, it has its own limitations, and the pathogenesis of this infection in hamsters differs from that in humans. In *Syrian hamsters*, the infection typically manifests benignly, similar to the majority of human cases, without progressing to the severe stages associated with cytokine storms.

Moreover, the progression of infection in this experimental model depends on numerous factors, including viral stock and load, as well as the hamster age, weight etc. These factors introduce complexities in synchronizing infection across different experimental groups and independent experiments. For instance, in our experiments, we utilized a single stock of the virus, and viral titers were determined immediately after stock preparation. We conducted two independent experiments with a 3-week interval. Despite using animals of similar weight and age, they were sourced from different batches. This likely explains some differences in the obtained results. For example, in the first experiment, the viral control group exhibited the greatest weight loss on day 7 after infection, reaching up to 5%. However, due to the variation in animal batches, we did not observe weight loss in the virus control group, despite using the same infection dose of 3.5 lgTCID_50_/mL.

For the coronavirus infection, “Immeran” may act on several stages, stimulating lung regeneration, activating antiviral immunity, and reducing pro-inflammatory cytokines that cause tissue damage and impact immune response. However, further analysis of molecular mechanisms of action is needed.

## 4. Materials and Methods

### 4.1. Animals

Female Golden *Syrian hamsters* 50–80 g (*Mesocricetus auratus*) were obtained from the Nursery for Laboratory Animals (IBCH RAS, Pushchino, Russia). Animals were housed individually at the Advisory Committee on Dangerous Pathogens (ACDP) containment level 3. Access to food and water was at will and environmental enrichment was provided. All experimental work was conducted according to the FELASA standards (FELASA Guidelines and Recommendations) [34]. All work with animals was carried out in accordance with the requirements of the European Convention for the Protection of Vertebrate Animals Used by the Institute for Experimental and Other Scientific Purposes (Strasbourg, 18th March1986). The research protocol was approved by the Research Ethics Review Committee of the I. I. Mechnikov Research Institute for Vaccines and Sera, Moscow, Russia (approval number: 02; from 14 October 2022).

### 4.2. Cells and Viruses

Vero CCL-81 cells were used to study antiviral activity against SARS-CoV-2 virus (“Dubrovka” Wuhan-like strain, identification number in GenBank: MW161041.1). Vero CCL-81 cells were purchased from ATCC (American Type Culture Collection). Cell lines were grown at 37 °C, with 5% CO_2_, in DMEM (Pan Eko, Moscow, Russia) supplemented with 5% heat-inactivated fetal bovine serum (FBS, Invitrogen, Carlsbad, CA, USA); 4.5 g/L of glucose (Sigma-Aldrich, St. Louis, MO, USA); 300 μg/mL of L-glutamine (Pan Eko, Moscow, Russia); and 40 μg/mL gentamicin (Sigma-Aldrich, St. Louis, MO, USA).

SARS-CoV-2 virus (“Dubrovka” Wuhan-like strain, GenBank: MW161041.1, phylogenetically similar to the Wuhan strain) was isolated from the nasopharynx aspirate and throat swab of the confirmed COVID-19 patients in Vero CCL-81 cells. Viral stocks (22.4 lgTCID_50_/mL) were prepared after serial passages in Vero CCL-81 cells using ‘infection media’ (detailed above). For all viruses, stocks were kept as aliquots at −80 °C.

SARS-CoV-2 virus (‘LIA’ strain, GenBank: ON032858.1) was isolated from the patient on 27 January 2022 (classification according to Pango BA.1.15, phylogenetically similar to the Omicron strain). In the experiment, a dose of 4.5 lgTCID_50_/mL was used.

### 4.3. Compounds

“Immeran” is a heteropolysaccharide with a molecular weight of about 70 kDa. It consists mainly of monosaccharides such as glucose, arabinose, and galactose, as well as xylose, mannose and galacto- and glucuronic acid residues in trace quantities. This heterogeneous carbohydrate structure classifies “Immeran” as a heteropolysaccharide. The presence of uronic acids indicates that acidic groups are also a component of this polymeric molecule. Dosages and route of administration were studied in preclinical and laboratory studies.

*Solanum tuberosum* sprouts extract solution (0.5 mg/mL) for intravenous injections (“Immeran”^®^, LLC Technopark-Center, Moscow, Russia) was used (Registration No LP-004128 from 8 February 17). The Favipiravir (Registration No: LP-006225 from 29.05.20) tablet was ground to a powder state. Powder was diluted with 200 mL of distilled water and given to the hamster orally.

### 4.4. Infection of Animals

All animals in the experiments were infected with the SARS-CoV-2 “Dubrovka” Wuhan-like strain virus intranasally under ether anesthesia in a volume of 100 μL per both nostrils. In both experiments, an infection dose 0.1 mL of 3.5 lgTCID_50_/mL was used. The animals were monitored daily throughout the experiment. The effectiveness of the drug in the infection model was assessed by the titer of the virus in the lungs of the animals on the 5th day of the experiment and the decrease in weight loss in the groups of treated animals was compared to the control group. Animals included in the comparison groups were intranasally administered phosphate-buffered saline in the same way and in the same volume.

Additionally, in the first experiment, the effectiveness was assessed by improving the condition of the lungs in pathomorphological studies; in the second experiment, neutralization reaction was additionally performed at the end of the experiment. The decrease or increase in animal weight was calculated separately for each hamster and expressed as a percentage. In this case, the weight of the animal before infection was taken as 100%. For all the hamsters in one group, the average percentage of weight loss or gain was determined.

### 4.5. Experimental Design

There were two independent series of experiments (the experiments flowchart is given in the Appendix A). In the first series of experiments, the therapeutical activity of “Immeran” was studied by assessing the influence of the treatment on the virus titer in lung tissue.

Animals were separated into six groups (Table 3), five animals each.

After five days, the animals were euthanized after anesthesia and the lungs were grouped for evaluation of virus titer and histology.

In the second series of the experiments, the effectiveness of the dose and regiment of treatment by “Immeran” was assessed.

Animals were separated into six groups (Table 4), eight animals each.

Five animals from each group were used further for the neutralization reaction and three for the evaluation of the virus titer in the lungs. Blood samples for the neutralization reaction were collected on day 21 after infection. At the same time, hamsters were weighed and the dynamics of the animals’ mass were assessed.

### 4.6. Determination of the Infectious Titer of the Virus in the Lungs

On day 5 after infection, hamsters in each group were euthanized under anesthesia and the lungs were removed under sterile conditions. After washing three times in a solution of 0.01 M PBS, the right lung was placed in a formaldehyde solution for histological examination (experiment 1). The left lung was homogenized and resuspended in 1 mL of cold sterile PBS solution. The suspension was cleared of cell debris by centrifugation at 2000× *g* for 10 min, and the supernatant was used to determine the infectious titer of the virus in cell culture.

To determine the infectious titer of the virus in hamster lungs, Vero CCL-81 cells were seeded in 96-well Costar plates with an average density of 20,000 cells per well and grown in DMEM with the addition of 5% fetal bovine serum, 10 mM glutamine and antibiotics (penicillin 100 IU/mL and streptomycin 100 µg/mL) until a complete monolayer was formed (within 3 days). Before virus infection, the cell culture was washed 2 times with DMEM without serum. We prepared 10-fold dilutions of each virus sample from the lungs from 10^−1^ to 10^−7^. The prepared dilutions in a volume of 200 μL were added to cell culture plates and incubated in an atmosphere of 5% CO_2_ at 37 °C for 5 days until the cytopathic effect (CPE) appeared in viral control cells. The result of the manifestation of CPE in cells was taken into account using the quantitative MTT test. The virus titer was calculated using the Ramakrishnan M.A formula in Excel and expressed in lgTCID_50_/mL. The average titer value was calculated for samples from hamsters of one group. Suppression of a viral titer twice or more lgTCID_50_/mL was considered reliable.

### 4.7. Neutralization Reaction

Blood from euthanized animals was collected in tubes without an anticoagulant. To form a clot, blood samples were kept in a thermostat at 37 °C for 30 min. After cutting off the serum, the blood was centrifuged two times for 15 min at 2000× *g*. The serum was collected and used to perform a neutralization reaction in cell culture. Each serum from each hamster was tested individually.

Vero CCL-81 cells were seeded into 96-well flat-bottomed culture plates, 12,000 cells/well, in a volume of 100 μL of freshly prepared MEM medium with 10% FBS and cultured for 24 h at a temperature of 37 °C in an atmosphere of 5% CO_2_ until a complete monolayer was formed. In this study, the human coronavirus strain SARS-CoV-2 was used, with an infectious activity of 6 lgTCID_50_/mL for Vero CCL-81 cells. Working solution (WS–MEM with glutamine and 2% FBS) was added to two-fold serial dilutions of hamster serum samples in a volume of 50 μL/well, a viral suspension was added at a dose of 4 lgTCID/mL in a volume of 50 μL/well, prepared on MS. The serum–virus mixture was incubated for 60 min. at 37 °C in an atmosphere of 5% CO_2_. Then, each sample prepared in this way was added to plates with a Vero-E6 cell culture, previously washed out once with serum-free medium. The cell culture with samples was incubated for 2 h at 37 °C in an atmosphere of 5% CO_2_. After incubation, the cells were washed out once with the medium and WS was added in a volume of 100 μL. The plates were then incubated at 37 °C in an atmosphere of 5% CO_2_ for 96 h. The results were taken into account after 72–96 h according to the cytopathic effect (CPE), when titration of the virus dose showed 50% or more CPE in the virus dose 1TCID_50_. Each serum sample was examined separately. Titer higher 1:10 was considered as positive.

### 4.8. Pathology Study

Extracted lung tissues were fixed in 10% formalin in PBS and processed for paraffin embedding. The paraffin blocks were cut into 3 mm-thick sections and then mounted on silane-coated glass slides. Each tissue sample was stained using a standard hematoxylin and eosin procedure. Histological pictures were taken at magnification ×40.

Histological sections of the lungs scored blindly for lung damage, using a cumulative severity score of 0 to 3 each for the most impotent parameters, as recently suggested (Gruber et al., 2020) [35], i.e., inflammation prevalence; extensive airless foci of pneumonia; small, isolated foci of pneumonia; bronchitis and bronchiolitis. The maximum possible score was 12 per animal. The raw data is presented in the Appendix A (Cumulative severity scoring from H&E stained slides of lungs from golden hamsters with SARS-CoV-2-associated interstitial pneumonia and different treatment dosages and regimes using the “Immeran” drug, Appendix A).

### 4.9. Data Analysis/Statistics

Data analysis was performed using the R-programming language in R-Studio. The dataset was first loaded into R-Studio, after which normality was assessed by constructing qq-plots and using the Shapiro–Wilk test (for n < 50), Lilliefors test considering sampling distribution parameters, and D’Agostino’s K-squared test. Depending on the normality test results, the Student’s *t*-test (for normally distributed data) or non-parametric Mann–Whitney U test (for non-normal data) were used for between-group comparisons. Bonferroni correction was applied for all multiple comparisons to adjust the α level. Virus titers were calculated using the Ramakrishnan M.A. method and expressed as lgTCID_50_/mL (given in Appendix A). Mean titers were calculated for samples from hamsters within each experimental group. A priori power analysis showed a total sample size of n = 35 was required to achieve 80% power for detecting significant treatment differences with α = 0.05 and expected effect size of 0.8 standard deviations. Post-hoc power for the weight loss outcome was 78%, indicating the study had acceptable power given the observed statistically significant results. All tests were two-sided with actual *p*-values < 0.05 considered statistically significant after Bonferroni adjustment.

## 5. Conclusions

Experimental data have demonstrated that “Immeran” is effective in prophylaxis and treatment of the SARS-CoV-2 infection, while being well-tolerated according to previous clinical studies. Based on the promising results of preclinical trials and preliminary data from ongoing clinical trials in COVID-19 patients, “Immeran” could possibly play a role in the complex treatment of COVID-19. However, it is still necessary to carry out a proper clinical trial.

## Figures and Tables

**Figure 1 viruses-16-00423-f001:**
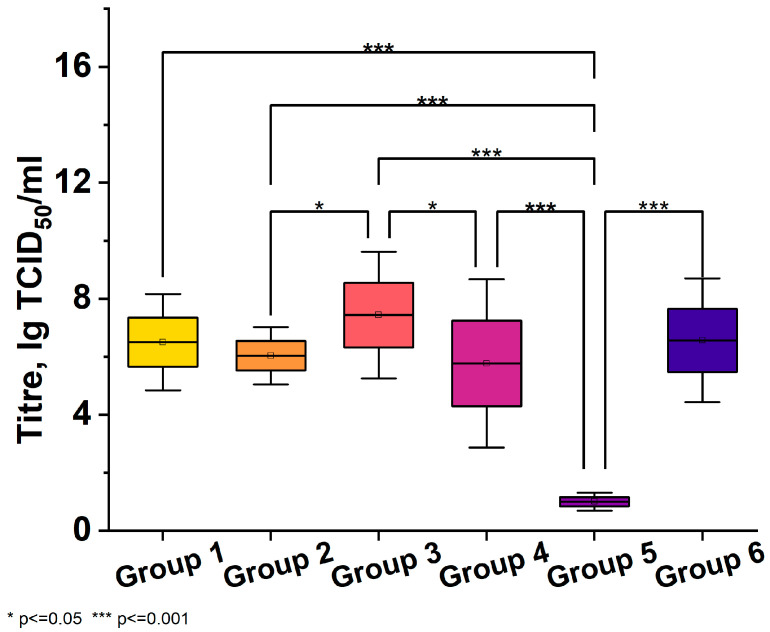
The figure shows viral titers in lung tissue 5 days after treatment in mice infected with the virus. Six groups were included: Group 1—Mice received an intraperitoneal injection of 0.5 mL physiological solution 1 h after infection and daily for 4 days. Group 2—Mice received one intraperitoneal injection of 0.5 mL of 250 µg/kg of “Immeran” 24 h before infection. Group 3—Mice received one intraperitoneal injection of 0.5 mL of 500 µg/kg of “Immeran” 24 h before infection. Group 4—Mice received one intraperitoneal injection of 0.5 mL of 250 µg/kg of “Immeran” 1 h after infection and daily for 4 days. Group 5—Mice received one intraperitoneal injection of 0.5 mL of 500 µg/kg of “Immeran” 1 h after infection and daily for 4 days. Group 6—Mice received Favipiravir orally at 1200 mg/kg 1 h before infection, then twice daily for 4 days. (n = 35). Viral titers were significantly reduced in the “Immeran” treatment groups compared to the viral control group, with the greatest effect observed at the 500 µg/kg dose administered 1 h after infection and for 4 days (*p* < 0.05, Kruskal-Wallis). Analysis was performed using Bonferroni adjustment for multiple comparisons.

**Figure 2 viruses-16-00423-f002:**
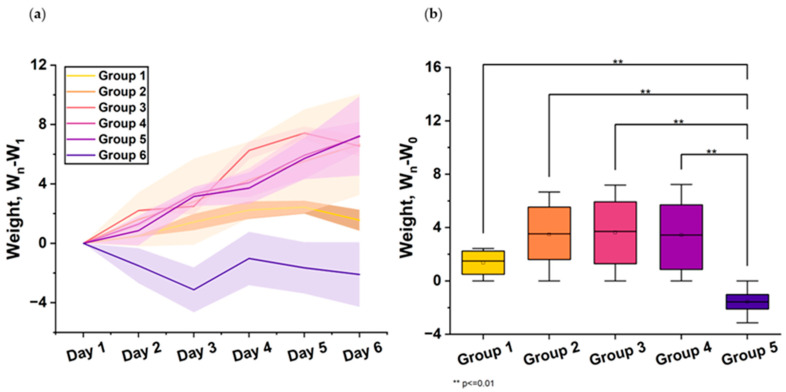
The figure shows body weight changes in mice across 6 treatment groups 6 days post-infection: Group 1—Viral control mice injected with physiological solution. Group 2—Mice treated with 250 μg/kg “Immeran” before infection. Group 3—Mice treated with 500 μg/kg “Immeran” before infection. Group 4—Mice treated with 250 μg/kg “Immeran” after infection. Group 5—Mice treated with 500 μg/kg “Immeran” after infection. Group 6—Mice treated with Favipiravir. (**a**) Changes in body weight of *Syrian hamsters* infected with SARS-CoV-2 in dependence on the treatment regimen in the first series of the experiments. The weight of the animal before infection was taken as 100%. (**b**) Daily animal weight measurements were normalized, with baseline weight (W_1_) minus the weight measured daily (W_n_). Statistical comparisons between groups were performed using unpaired Student’s *t*-test (*p* < 0.05, n = 35) with Bonferroni adjustment for multiple comparisons. Error bars denote 95% confidence intervals calculated using Student’s t-distribution, indicating the likely range of true mean weight change.

**Figure 3 viruses-16-00423-f003:**
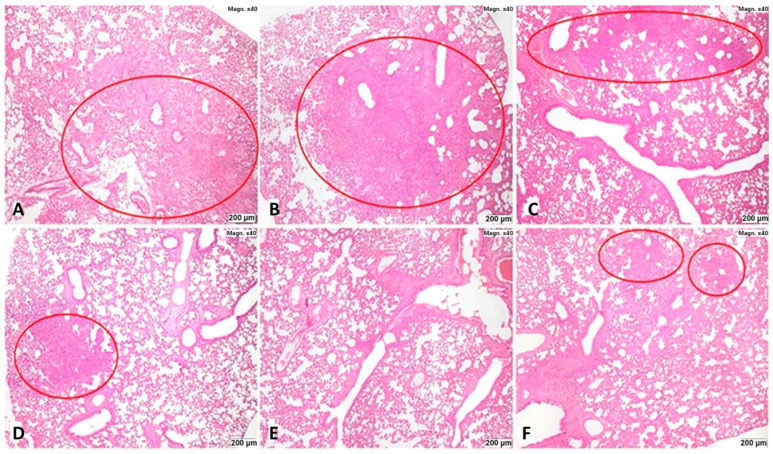
Histopathology of hamster lungs on the 4th day after infection with SARS-CoV-2: (**A**)—group 1 (viral control group), (**B**)—group 2–250 μg/kg of “Immeran” 24 h before infection, (**C**)—group 3–500 μg/kg of “Immeran” 24 h before infection, (**D**)—group 4–250 μg/kg of “Immeran” 1 h after infection and four days once every other day, (**E**)—group 5–500 μg/kg of “Immeran” 1 h after infection and four days once every other day, (**F**)—group 6 (favipiravir control). Foci of interstitial pneumonia in the respiratory lobe of the right lung (red ovals). Magnification ×40. Hematoxylin and eosin staining. The statistical analysis for the table is presented in the Appendix A.

**Figure 4 viruses-16-00423-f004:**
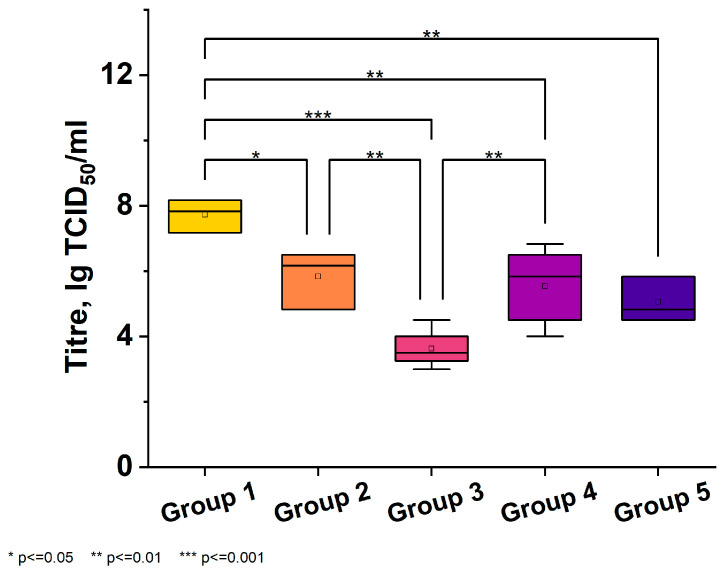
The figure shows viral titers in lung tissue 5 days after treatment in mice infected with the SARS-CoV-2 virus. Five groups were included: Group 1—Viral control-mice received intraperitoneal injection of physiological solution 1 h after infection and daily for 4 days. Group 2—Mice received one intraperitoneal injection of 1000 μg/kg “Immeran” 24 h before infection. Group 3—Mice received one intraperitoneal injection of 500 μg/kg “Immeran” 1 h after infection and daily for 4 days. Group 4—Mice received one intraperitoneal injection of 500 μg/kg “Immeran” 1 h after infection and once every other day for 4 days. Group 5—Mice received one intraperitoneal injection of 1000 μg/kg “Immeran” 1 h after infection and once every other day for 4 days. Viral titers were significantly reduced in the “Immeran” treatment groups compared to viral controls. The greatest reduction was seen in Group 5 treated with 1000 μg/kg “Immeran” at 1 h post-infection and every other day (*p* < 0.05, Kruskal–Wallis test, n = 35).

**Figure 5 viruses-16-00423-f005:**
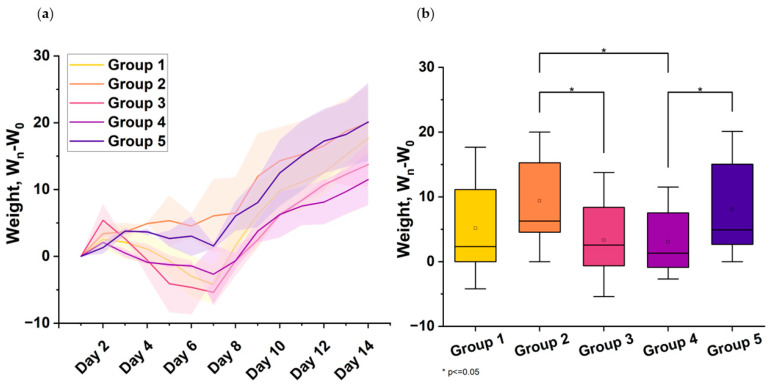
The figure shows body weight changes in infected mice across 5 treatment groups 14 days post-infection: Group 1—Viral control—mice injected with physiological solution. Group 2—Mice treated with 1000 μg/kg “Immeran” before infection. Group 3—Mice treated with 500 μg/kg “Immeran” after infection daily. Group 4—Mice treated with 500 μg/kg “Immeran” after infection every other day. Group 5—Mice treated with 1000 μg/kg “Immeran” after infection every other day. (**a**) Changes in body weight of *Syrian hamsters* infected with SARS-CoV-2 in dependence on the treatment regimen in the second series of experiments. The weight of the animal before infection was taken as 100%. (**b**) Animal daily weight measurements were normalized, with baseline weight (W_0_) minus the weight measured daily (W_n_). Statistical comparisons between groups were performed using unpaired Student’s *t*-test (*p* < 0.05, n = 35) with Bonferroni adjustment for multiple comparisons. Error bars denote 95% confidence intervals calculated using Student’s t-distribution, indicating the likely range of true mean weight change.

**Table 1 viruses-16-00423-t001:** Morphological picture of SARS-CoV-2-associated interstitial pneumonia in golden hamsters with different treatment dosage and regimes using “Immeran” drug.

Inflammation Severity Criteria	Groups (n = 5)					
	1 Viral Control	2 Experiment 250 μg/kg of “Immeran” 24 h before Infection	3 Experiment 500 μg/kg of “Immeran” 24 h before Infection	4 Experiment 250 μg/kg of “Immeran” 1 h after Infection and Four Days Once Every Other Day	5 Experiment 500 μg/kg of “Immeran” 1 h after Infection and Four Days Once Every Other Day	6 Favipiravir Control 1200 mg/kg 1 h before Infection, Then 500 mg/kg Twice a Day for 4 Days
Number of animals without inflammation in lungs	0	0	1	2	3	0
Inflammation prevalence	in 3–4 lung lobes	in 1–2 lung lobes	in 1–2 lung lobes	in 1–2 lung lobes	in 1 lung lobe	in 1–2 lung lobes
Extensive airless foci of pneumonia	+	+	-	-	In one animal in 1 lung lobe	-
Small, isolated foci of pneumonia	-	-	+	+	+	+
Bronchitis and -olitis	+	+	+	-	-	+

**Table 2 viruses-16-00423-t002:** The effect of drug treatment on the expression of virus-neutralizing antibodies in *Syrian hamsters* infected with SARS-CoV-2.

	Animal	Titer in Neutralization Reaction	
		SARS-CoV-2	
		“Dubrovka” Wuhan-like Strain	Omicron Strain
Group 1 viral control 0.5 mL of physiological solution injected intraperitoneally 1 h after infection and four days daily	A	1:80	1:160
	B	1:40	1:320
	C	1:40	1:160
	Mean	1:53	1:213
Group 2 1000 μg/kg of “Immeran” 24 h before infection	A	1:160	1:320
	B	1:80	1:160
	C	1:80	1:160
	Mean	1:106	1:213
Group 3 500 μg/kg of “Immeran” 1 h after infection and four days daily	A	1:320	1:640
	B	1:320	1:320
	C	1:160	1:320
	Mean	1:267	1:426
Group 4 500 μg/kg of “Immeran»” 1 h after infection and four days once every other day	A	1:80	1:160
	B	1:80	1:320
	C	1:80	1:160
	Mean	1:80	1:213
Group 5 1000 μg/kg of “Immeran” 1 h after infection and four days once every other day	A	1:160	1:320
	B	1:80	1:160
	C	1:160	1:320
	Mean	1:133	1:266
Group 6 0.5 mL of physiological solution injected intraperitoneally 1 h after infection and four days daily	A	˂1:10	˂1:10
	B	˂1:10	˂1:10
	C	˂1:10	˂1:10
	Mean	˂1:10	˂1:10

**Table 3 viruses-16-00423-t003:** Experimental and control groups of the first experimental series.

Group	Dose and Regiment
1 viral control group	0.5 mL of physiological solution injected intraperitoneally 1 h after infection and four days daily
2 experimental group	one intraperitoneal injection 0.5 mL of 250 μg/kg of “Immeran” 24 h before infection
3 experimental group	one intraperitoneal injection 0.5 mL of 500 μg/kg of “Immeran” 24 h before infection
4 experimental group	one intraperitoneal injection 0.5 mL of 250 μg/kg of “Immeran” 1 h after infection and four days once every other day
5 experimental group	one intraperitoneal injection 0.5 mL of 500 μg/kg of “Immeran” 1 h after infection and four days once every other day
6 Favipiravir group	Favipiravir orally 1200 mg/kg 1 h before infection, then 500 mg/kg twice a day for 4 days

**Table 4 viruses-16-00423-t004:** Experimental and control groups of the second experimental series.

Group	Dose and Regiment
1 viral control group	0.5 mL of physiological solution injected intraperitoneally 1 h after infection and four days daily
2 experimental group	one intraperitoneal injection 0.5 mL of 1000 μg/kg of “Immeran” 24 h before infection
3 experimental group	one intraperitoneal injection 0.5 mL of 500 μg/kg of “Immeran” 1 h after infection and four days daily
4 experimental group	one intraperitoneal injection 0.5 mL of 500 μg/kg of “Immeran” 1 h after infection and four days once every other day
5 experimental group	one intraperitoneal injection 0.5 mL of 1000 μg/kg of “Immeran” 1 h after infection and four days once every other day
6 uninfected control group	0.5 mL of physiological solution injected intraperitoneally 1 h after infection and four days daily

## Data Availability

The data presented in this study are available on request from the corresponding author.

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
