# Peer review of "Evaluation of the Polysaccharide “Immeran” Activity in Syrian hamsters’ Model of SARS-CoV-2"

_viruses, 2024, doi:10.3390/v16030423_

Round 1

Reviewer 1 Report

Comments and Suggestions for Authors

The document provided is an experimental study on the antiviral activity of the polysaccharide "Immeran" against SARS-CoV-2 in Syrian hamsters. The paper has some merit however it could be improved according to some minor revisions.

To enhance the quality and impact of the paper on "Immeran" activity against SARS-CoV-2 in Syrian hamsters, consider the following suggestions:

Expand on Statistical Details: Provide more detailed statistical analysis information, including exact p-values, confidence intervals, and the statistical power of the study.

Include Limitations and Potential Biases: Clearly outline any limitations of the study, including those inherent in the animal model used, and discuss any potential biases in study design or execution.

Discuss Human Relevance: While acknowledging the need for clinical trials, offering a more detailed discussion on the translation of these findings from animal models to potential human treatment could be valuable. This might include comparing the "Immeran" doses used in hamsters with equivalent doses for humans, based on body surface area conversion, and discussing any known safety data in humans and how could be potentially of use.

Mechanism of Action: If possible, include more information or hypotheses about the mechanism of action of "Immeran".

Enhance the Discussion on Public Health Implications: Expand the discussion on the potential public health implications of "Immeran", especially in the context of the ongoing search for effective COVID-19 treatments. Addressing how "Immeran" could fit into existing treatment protocols or strategies for managing COVID-19 could make the paper more relevant to a broader audience.

Incorporating these suggestions can increase the paper's transparency, robustness, and relevance, potentially making it a more valuable contribution to the field of antiviral research.

Author Response

Good day honorable sir/madam!

Thank you very much for valuable comments. Please see attached file with answers. Also, it is in need, to state that yet we cannot attach revised manuscript and supplementary materials, due to limitations of the system. However, we have sent our revised files to the editor, so he can share them with reviewers.

Best regards, Evgenii Generalov

Reviewer 2 Report

Comments and Suggestions for Authors

The study is interesting, and the results may have a potential interest in medicine and the pharmacological industry.

I suggest a revision of the manuscript. My comments are below.

1.      Most cited publications are new and appropriate; only one should be corrected: lines 64–66. The authors mentioned the antiviral activity of polysaccharides from Chaga, but the cited paper is about the antiviral and antimicrobial activities of carrageenan, a polysaccharide of red marine algae. The citation should be replaced with an appropriate one, and it is also recommended to not cite such obsolete works (1987) and substitute newer ones.

2.      Line 89-90. The figure captures Probably, the detailed description of all “Immeran” experimental groups can be added to the capture (as a footnote or comment, or just extend the description of Figure 1 below the fig.) in order to better understand the obtained results without the necessity of going to the chapter “Material and Methods.”

3.      Line 101: “The use of favipiravir (Group 6) was not effective and affecting the virus…” The formulation is not the best; "affecting" may mean an increase as well as a decrease in virus titer. It probably might be changed to something clearer, like “lead to an increase in virus titer or result in an increase.”

4.      Line 108. I would recommend replacing the strong “animals were killed” with “animals were euthanized.

5.      Line 147-148. Repetition: “were noted. It should be noted…”. Re-formulation is recommended.

6.      Line 160. All the time through the whole text, the authors use the style "mcg."  Here suddenly appears “1000 μg/kg." The entire text check is recommended to standardize the usage of units.

7.      Line 190. In the text, “Wuhan strain," but in the table below, “Dubrovka strain." Additional clarification is recommended without necessity to look for information in “Material and Methods” or “Discussion” (line 252: CoV-2 “Dubrovka Wuhan-like strain") to determine if the same strain is meant or something different.

8.      Line 201: “1000 mcg/ml...” and line 202: “500 and 1000 μg/ml...”.

9.      Line 223-227. Discussion. Why is the discussion chapter starting with a focus on the immunomodulatory activity of polysaccharides? In the current work, the antiviral effect of the polysaccharide drug is stated. It is advised to reconstruct the discussion chapter and mention only polysaccharide effects that are related to the current study or interrelate (and explain how and why) the mentioned ones with the effects described in the current study.

10.  Line 251-256. The formulation is not good, and it is very difficult to understand the meaning of the sentences (explanation of previously described antiviral effects of “Immeran”). Particularly, the sentence starts with the statement, “In previous preliminary experiments, we found that Immeran had no antiviral activity against SARS-CoV-2.” Then, it followed the more detailed description of the “Immeran” antiviral actions “in the case of coronavirus infection." It should be reconstructed to properly present the results mentioned in this part.

11.  Line 260. “…its reparatory characteristics could be useful for lung tissue damage.” Probably “be useful in the treatment of?” It is recommended to rephrase the sentence.

12.  Line 264. It should be summarized from the current and previous experimental series how the effect of “Immeran” on COVID infections was demonstrated.

13.  Line 265. There is no data about the influence of “Immeran” on cytokine production in the current work.

14.  A major reconstruction of the discussion part is required. Please revise the whole content of the section. It should be accurate and clear to understand. All the papers discussed and previously obtained results should be discussed in direct connection with the results of the current work.

15.  Line 333, line 365. Replace “animals were humanely killed” with “animals were euthanized.”

16.  Editing language is advised.

Author Response

(The authors gave the same response as above.)

Reviewer 3 Report

Comments and Suggestions for Authors

Thanks to the authors for providing this manuscript. The research focused on exploring the antiviral properties of the drug "Immeran" to SARS-CoV-2. It is attractive that virus titer was suppressed post therapy in the animal model used in this study.

Major points:

1.      The grouping design of the two animal experiments looks very confusing. Please add flowcharts based on animal experiments 1 and 2 to better understand the experimental design.

2.      For the immunohistochemical experiment in Figure 3, please add and present the statistical analysis of the area or intensity of the lung parenchyma with inflammation in each experimental group, instead of representative images. Similarly, please statistically analyze the difference in titer of virus-neutralizing antibodies shown in table 2.

3.      “During the experiment, until animals were killed, there was no weight loss ob- served in «Immeran» and viral control groups” mentioned in line 108-109. “In the second series of the experiments, weight loss in the viral control group (Group 1) was greatest on day 7 after infection, reaching up to 5%.” Mentioned in line 169-170. Author also mentioned that “In both experiments, an infection dose of 103.5TCID50 was used” (line 313-314) But there is a contradiction in the weight changes of the virus control group in the two animal experiments. Please explain why this difference occurs.

Minor points:

1.      Please use consistent units throughout the entire text, as some parts use mcg/ml and others use μg/ml.

2.      The symbol « » to represent a drug seems a bit unfamiliar to me. I am not sure if this way of representation is appropriate.

3.      What does the positive control group represent (line 131)? I did not find other description of positive controls in the entire manuscript except for here. Please correct me if I miss any information.

4.      Is the format of the tables in the manuscript appropriate? They do not appear to be in a strict three-line table format.

5.      The figure legends in the manuscript are too brief, especially the legend of figure 4. Generally, the experimental design, data, statistical methods, animal numbers, etc. need to be basically described in the figure legend.

Author Response

Good day honorable sir/madam!

Please see attached file with answers to your comments.

We are unable to attache supplement material and revised version of the manuscript. However, we have sent it to the editor, so he can share it with reviewers and use in futher publication activity,

Best regards, Evgenii Generalov

Reviewer 4 Report

Comments and Suggestions for Authors

In this work, «Immeran» is demonstrated to be effective in prophylaxis and treatment of SARS-CoV-2 infection, while being well tolerated according to previous clinical studies. The work is important and interesting. I think it can be published after minor revision.

All of ml need change to mL.

Table 1 and Table 2 need to be changed to a three line grid.

Figures 2 and 5 have unclear meanings and require explanation. What does Wn-W1 represent?

It is difficult to speculate on the possible action mechanism of «Immeran» based on existing data, and more molecular biology experimental data is needed to support the speculation of the mechanism.

Pay attention to the reference format, there are many issues that need to be uniformly changed.

I don't know why, but I can't see any supporting information in the review system.

Comments on the Quality of English Language

can be improved

Author Response

Good day honorable sir/madam!

Please see attached file with answers. 

We also do not have an ability to attach a revised manuscript to the system.

Best regards, GEA

Round 2

Reviewer 3 Report

Comments and Suggestions for Authors

Thanks for the author's careful revision of the manuscript, which has improved its quality. There is still minor issue that need to be addressed. Please revise the following issues and their similar issues in the manuscript.

Minor points:

1.      The figures and tables in the supplementary file should be cited in the paragraphs in the results section where they are mentioned. For example, XXXXXXXXX (Figure S1, Table S1……) XXXXXX.

2.      In the manuscript, there are multiple ways of indicating virus titers, such as 6.50±0.84 lgTCID50/mL, 104 TCID/mL, or 10^3.5 TCID50. Please ensure consistency by using the same format throughout the manuscript.

3.      In the line 396. “In the experiment a dose 104,5TCID50/mL was used”. What does the comma between the numbers 4 and 5 mean? Please verify.

Author Response

Good day honorable sir/madam!

Thank you for your comment! Everything corrected. Please see revised manuscript.

Best regards, Generalov Evgenii
